

# Total power calibration in FTIR emission spectroscopy for finite interferograms

Lukas Heizmann[1], Mathias Palm[1], Justus Notholt[1], and Matthias Buschmann[1]

[1]Universität Bremen, Bibliothekstraße 1, 28359 Bremen.

**Correspondence:** Lukas Heizmann (heizmann@iup.physik.uni-bremen.de)

**Abstract.** The commonly used total power calibration procedure in FTIR (Fourier transform infrared) emission spectroscopy first described by Revercomb et al. (1988) is strictly speaking only valid for double-sided, infinite interferograms. Here we investigate the effect of interferogram truncation on calibrated mid-resolution emission spectra, describe the underlying theory, and quantify the errors in a case study. This quantification is important as the demands on precision in atmospheric measure-
5 ments increase. While application of the Revercomb formula might lead to large errors in the unequal-sided case, we show that one can obtain the same spectral estimate as from equal-sided interferograms. This is achieved by modifying Revercomb's method incorporating a phase correction procedure using low-resolution phase spectra from the black body measurements. We include a case study simulating atmospheric and black body spectra with a line-by-line radiative transfer model. For a mid-resolution use case, we find that the effect of truncation on the spectral radiance measurement is generally well below 0.05 %.
Only in spectral regions where strong absorption lines in the black body spectra are present can the errors be larger. However, those spectral regions are usually saturated in atmospheric spectra for observations from the ground and thus contain little information about the atmosphere except for the lowermost layer and can and usually should be excluded for most retrieval procedures.

## 1 Introduction

FTIR (Fourier transform infrared) emission spectroscopy is an established tool in atmospheric remote sensing. It is used to validate radiative transfer calculations, to continuously measure the thermodynamic state of the atmosphere (temperature and water vapor content) and the radiative properties of clouds (Knuteson et al., 2004b). It has also been used in trace gas (Mariani et al., 2013) and aerosol retrievals (Ji et al., 2023). One of the well-known instruments commonly deployed for ground-based FTIR emission spectroscopy is the AERI (Atmospheric Emitted Radiance Interferometer, Knuteson et al. 2004b)
which measures down-welling radiation in the mid-infrared spectral range with a resolution of $1 \ \mathrm{cm}^{-1}$. It records equal-sided interferograms. In order to achieve higher resolution, which is especially important for trace gas retrievals, instruments such as the NYAEM-FTS (Ny-Ålesund emission Fourier transform spectrometer) are deployed in single-sided interferogram mode. The spectrometer used in this instrument is a commercial Vertex80 by Bruker. The maximum optical path difference for the long interferogram side is 11.25 cm and about 1 cm for low-resolution double-sided phase spectra. The instrument is used for
aerosol retrievals (Ji et al., 2023) and trace gas retrievals are currently being developed.





In emission spectroscopy, absolute radiance calibration is needed for most applications since, contrary to absorption spectroscopy, there is no broadband background which allows for retrieving most information from relative line intensities. Total power calibration using two reference black bodies at known temperature and emissivity is commonly performed. Revercomb et al. (1988) extended the simple linear interpolation calibration scheme in order to incorporate the internal infrared emission from the beam splitter of the instrument with its peculiar phase properties.

In Sect. 2 we review the linear interpolation scheme for total power calibration where two black body spectra at known temperature are used to calibrate the atmospheric spectrum. Those two additional spectra are needed to obtain the instrument's responsivity (a scaling factor) and the offset between radiance input to the instrument and electric output of the detector assuming a linear relationship at each wavenumber. In Sect. 3 we use infrared radiative transfer calculations to model the absorption lines in the black body spectra. In Sect. 4 we review the main idea by Revercomb et al. (1988) in order to incorporate phase errors in the calibration scheme. In Sect. 5 we apply equal- and unequal-sided truncation to the three interferograms involved in the calibration and investigate its effect on the calibrated spectrum, since the calibration procedure described by Revercomb et al. (1988) is strictly speaking only valid for infinite, equal-sided interferograms. We describe the underlying theory and quantify the effects in a case study. Generally, the errors might be larger when Revercomb's method is applied to unequal-sided interferograms compared to equal-sided truncation. By modifying Revercomb's method we present a means to obtain the same spectral estimate from unequal-sided interferograms as from equal-sided interferograms. This is achieved by incorporating a phase correction procedure using a low-resolution phase spectrum obtained from the black body measurements.

## 2 Total power calibration

When measuring infrared radiation in solar absorption geometry, the transmission of the atmosphere is often of interest, i. e. which fraction of the incoming solar radiation is absorbed by the atmospheric gases of interest. To this end we do not need to know the absolute radiance of the incoming radiation. A fit to the broadband background of the solar spectrum suffices. This is different when measuring emission spectra. Most applications like e. g. trace gas retrievals, retrievals of the thermodynamic profiles or aerosols require absolute radiance calibration. This is usually achieved in a process called total power calibration using two light sources with an emissivity close to 1 (black bodies) at different temperatures which are measured in a sequence with the actual atmospheric measurements. Since the emission of black bodies is known exactly by Planck's law, provided that the exact temperature of the black bodies and their wavenumber dependent emissivity is known, we can use linear interpolation between the hot and cold black body to calibrate the atmospheric signal. This procedure assumes a linear response of the instrument to the incoming radiance $S$, i. e. of the measured electric output signal $S_{\mathrm{m}}$ of the detector in the required energy range. The detector response can be described by:

$$S_{\mathrm{m}}(\nu) = r(\nu)[S(\nu) + S_{\mathrm{i}}(\nu)] \tag{1}$$

where $r$ the responsivity of the instrument, $S$ the incoming radiance and $S_{\mathrm{i}}$ the radiation emitted from the instrument itself, which is important in the thermal infrared spectral region that we consider. Since this relation holds for the unknown radiance of





an atmospheric scene $S_{\mathrm{s}}$, the hot black body and the cold black body for which the radiances $S_{\mathrm{h}}$ and $S_{\mathrm{c}}$ are known theoretically by the Planck law, we have three equations that we can solve for the three unknowns $S_{\mathrm{s}}$, $r$ and $S_{\mathrm{i}}$:

$$S_{\mathrm{m,n}}(\nu) = r(\nu)\left[S_{\mathrm{n}}(\nu) + S_{\mathrm{i}}(\nu)\right] \tag{2}$$

for $\mathrm{n} = \mathrm{s,c,h}$. So

$$\frac{S_{\mathrm{m,s}}(\nu) - S_{\mathrm{m,c}}(\nu)}{S_{\mathrm{m,h}}(\nu) - S_{\mathrm{m,c}}(\nu)} = \frac{S_{\mathrm{s}}(\nu) - S_{\mathrm{c}}(\nu)}{S_{\mathrm{h}}(\nu) - S_{\mathrm{c}}(\nu)} \tag{3}$$

and we can thus solve for $S_{\mathrm{s}}(\nu)$ to get the calibrated spectrum (Revercomb et al., 1988)

$$S_{\mathrm{s,calib}}(\nu) = \frac{S_{\mathrm{m,s}}(\nu) - S_{\mathrm{m,c}}(\nu)}{S_{\mathrm{m,h}}(\nu) - S_{\mathrm{m,c}}(\nu)}\left[S_{\mathrm{h}}(\nu) - S_{\mathrm{c}}(\nu)\right] + S_{\mathrm{c}}(\nu). \tag{4}$$

## 65   3   Black body spectra

While it is common to assume that the black body spectra are smooth functions of wave number, in reality they exhibit absorption lines from the gases, especially water vapor in the air column between black body and beam splitter of the spectrometer. Figure 1 shows a measured black body spectrum of the SR800 by CI Systems at 110 °C built into the NYAEM-FTS with characteristic absorption lines.

The observed black body spectrum can be calculated using the radiative transfer equation (Sinnhuber, 1999; Notholt et al., 2006) as the emission from the surface of the black body $L_{\mathrm{bb}}(\nu)$ times the transmittance $T_0(\nu)$ through the lab air layer between the black body and the spectrometer plus the emission of this layer $(1 - T_0)B_{T_{\mathrm{lab}}}$:

$$S_{\mathrm{bb}}(\nu) = L_{\mathrm{bb}}(\nu)T_0(\nu) + (1 - T_0(\nu))B_{T_{\mathrm{lab}}}(\nu) \tag{5}$$

The spectrum emitted from the surface of the black body $L_{\mathrm{bb}}(\nu)$ is a Planck spectrum $B_{T_{\mathrm{bb}}}(\nu)$ reduced (multiplied) by the
emissivity $\epsilon_{\mathrm{bb}}(\nu)$ plus the reflected emission from the surrounding (which we assume to be a black body as a first approximation), where the reflectance is given by $1 - \epsilon_{\mathrm{bb}}(\nu)$:

$$L_{\mathrm{bb}}(\nu) = \epsilon_{\mathrm{bb}}(\nu)B_{T_{\mathrm{bb}}}(\nu) + (1 - \epsilon_{\mathrm{bb}}(\nu))B_{T_{\mathrm{lab}}}(\nu). \tag{6}$$

And thus the observed spectrum is

$$S_{\mathrm{bb}}(\nu) = \left[\epsilon_{\mathrm{bb}}(\nu)B_{T_{\mathrm{bb}}}(\nu) + (1 - \epsilon_{\mathrm{bb}}(\nu))B_{T_{\mathrm{lab}}}(\nu)\right]T_0(\nu) + (1 - T_0(\nu))B_{T_{\mathrm{lab}}}(\nu). \tag{7}$$

Inserting this for the hot and cold black body spectra respectively in the calibration equation and from now on dropping the argument $\nu$ for better readability, we get

$$S_{\mathrm{s,calib}} = \frac{S_{\mathrm{m,s}} - S_{\mathrm{m,c}}}{S_{\mathrm{m,h}} - S_{\mathrm{m,c}}}\left[\epsilon_{\mathrm{h}}(B_{\mathrm{h}} - B_{\mathrm{lab}}) - \epsilon_{\mathrm{c}}(B_{\mathrm{c}} - B_{\mathrm{lab}})\right]T_0 + \left[\epsilon_{\mathrm{c}}B_{\mathrm{c}} + (1 - \epsilon_{\mathrm{c}})B_{\mathrm{lab}}\right]T_0 + (1 - T_0)B_{\mathrm{lab}}. \tag{8}$$





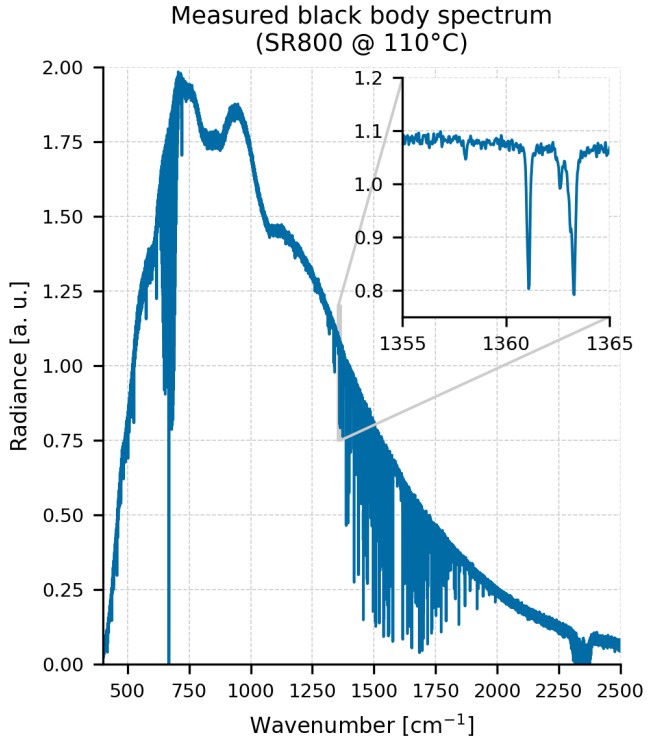

**Figure 1.** Hot black body spectrum containing lab air as observed by the NYAEM-FTS in Ny-Ålesund, Svalbard.

This is unfortunate, since it requires knowledge of $T_0$ for calibration, i. e. the entire state and composition of the lab air as well as the entire spectroscopic information on the absorption lines. However, if we realize that we can split the scene spectrum into

85 the actual atmospheric spectrum $S_\mathrm{a}$ and the layer between black body and instrument which itself absorbs and emits radiation

$$S_\mathrm{s} = S_\mathrm{a} T_0 + (1 - T_0) B_\mathrm{lab} \tag{9}$$

and insert this in the measured scene spectrum

$$
\begin{aligned}
\frac{S_\mathrm{s} - S_\mathrm{c}}{S_\mathrm{h} - S_\mathrm{c}} &= \frac{S_\mathrm{a} T_0 + \cancel{(1 - T_0) B_\mathrm{lab}} - [\epsilon_\mathrm{c} B_\mathrm{c} + (1 - \epsilon_\mathrm{c}) B_\mathrm{lab}] T_0 + \cancel{(1 - T_0) B_\mathrm{lab}}}{[\epsilon_\mathrm{h}(B_\mathrm{h} - B_\mathrm{lab}) - \epsilon_\mathrm{c}(B_\mathrm{c} - B_\mathrm{lab})] T_0} \\
&= \frac{[S_\mathrm{a} - \epsilon_\mathrm{c} B_\mathrm{c} - (1 - \epsilon_\mathrm{c}) B_\mathrm{lab}] \cancel{T_0}}{[\epsilon_\mathrm{h}(B_\mathrm{h} - B_\mathrm{lab}) - \epsilon_\mathrm{c}(B_\mathrm{c} - B_\mathrm{lab})] \cancel{T_0}} \\
&= \frac{[S_\mathrm{a} - L_\mathrm{c}] \cancel{T_0}}{[L_\mathrm{h} - L_\mathrm{c}] \cancel{T_0}}.
\end{aligned}
\tag{10}
$$





the emission from the lab air cancels in the difference and the absorption cancels in the ratio of the measured spectra. We can
thus obtain the calibrated atmospheric spectrum without knowing $T_0$ as

$$
\begin{aligned}
S_{\mathrm{a,calib}} &= \frac{S_{\mathrm{m,s}} - S_{\mathrm{m,c}}}{S_{\mathrm{m,h}} - S_{\mathrm{m,c}}} \left[ L_{\mathrm{h}} - L_{\mathrm{c}} \right] + L_{\mathrm{c}} \\
&= \frac{S_{\mathrm{m,s}} - S_{\mathrm{m,c}}}{S_{\mathrm{m,h}} - S_{\mathrm{m,c}}} \left[ \epsilon_{\mathrm{h}}(B_{\mathrm{h}} - B_{\mathrm{lab}}) - \epsilon_{\mathrm{c}}(B_{\mathrm{c}} - B_{\mathrm{lab}}) \right] + \left[ \epsilon_{\mathrm{c}} B_{\mathrm{c}} + (1 - \epsilon_{\mathrm{c}}) B_{\mathrm{lab}} \right].
\end{aligned}
\tag{11}
$$

So we can basically use eq. 4 and insert for the black body spectra the spectra $L_{\mathrm{bb}}$ which we would observe if there was no absorption and emission by the lab air, just the black body emission and the reflection of the surrounding. A further simplification arises if the cold black body is at room temperature:

$$
S_{\mathrm{a,calib}} = \frac{S_{\mathrm{m,s}} - S_{\mathrm{m,c}}}{S_{\mathrm{m,h}} - S_{\mathrm{m,c}}} \left[ \epsilon_{\mathrm{h}}(B_{\mathrm{h}} - B_{\mathrm{c}}) \right] + B_{\mathrm{c}}.
\tag{12}
$$

## 4 Phase errors

In Fourier transform spectroscopy real world interferograms are not completely symmetric, which can be due to a variety of causes (Davis et al., 2001): Dispersion in the instrument leads to a wavelength-dependent path difference, so the zero path difference (ZPD) can be at different positions for every wavelength and thus cannot be well defined. If sampling points of the interferogram are chosen such that there is no sampling point exactly at the position of the ZPD (even if it is well defined) the interferogram appears asymmetric. If the source intensity varies during the recording of the interferogram, the interferogram is also asymmetric. While the Fourier transform of symmetric functions is real, the Fourier transform of asymmetric interferograms leads to complex spectra. This phenomenon is called phase error in FTIR spectroscopy. In FTIR emission spectroscopy, Revercomb et al. (1988) describe that the contribution to the signal from the instrument's infrared radiation originating at the beam splitter (internal) might have a different phase error than the signal of the measured source itself, i. e. there is a phase shift $\phi_{\mathrm{i}}(\nu)$ relative to the source signal (external) $\phi_{\mathrm{s}}(\nu)$ and therefore amplitude spectra cannot be used because of artifacts due to complex contribution.

The measured complex spectrum has the following form:

$$
S_{\mathrm{m}}(\nu) = r(\nu) \left[ S(\nu) + S_{\mathrm{i}}(\nu) \mathrm{e}^{\mathrm{i}\phi_{\mathrm{i}}(\nu)} \right] \mathrm{e}^{\mathrm{i}\phi_{\mathrm{s}}(\nu)} = \int_{-\infty}^{\infty} I(x) \mathrm{e}^{-\mathrm{i}2\pi\nu x} \mathrm{d}x.
\tag{13}
$$

Revercomb et al. proposed that if one uses the entire complex spectra in the same calibration equation, assuming that the external phases are the same for all involved measurements (hot, cold, atmosphere), the internal phases cancel in the subtraction and the external phases cancel in the quotient. Since the Fourier transform is linear, one can also take the difference of the




interferograms and then apply the Fourier transform, provided that the ZPD of the interferograms is at the same position.

$$
\begin{aligned}
S_{\mathrm{s,calib}}(\nu) &= \frac{S_{\mathrm{m,s}}(\nu) - S_{\mathrm{m,c}}(\nu)}{S_{\mathrm{m,h}}(\nu) - S_{\mathrm{m,c}}(\nu)}\left[S_{\mathrm{h}}(\nu) - S_{\mathrm{c}}(\nu)\right] + S_{\mathrm{c}}(\nu)\\
&= \frac{\cancel{r(\nu)}\left[S_{\mathrm{s}}(\nu) + \cancel{S_{\mathrm{i}}(\nu)\mathrm{e}^{\mathrm{i}\phi_{\mathrm{i}}(\nu)}} - S_{\mathrm{c}}(\nu) - \cancel{S_{\mathrm{i}}(\nu)\mathrm{e}^{\mathrm{i}\phi_{\mathrm{i}}(\nu)}}\right]\cancel{\mathrm{e}^{\mathrm{i}\phi_{\mathrm{s}}(\nu)}}}{\cancel{r(\nu)}\left[S_{\mathrm{h}}(\nu) + \cancel{S_{\mathrm{i}}(\nu)\mathrm{e}^{\mathrm{i}\phi_{\mathrm{i}}(\nu)}} - S_{\mathrm{c}}(\nu) - \cancel{S_{\mathrm{i}}(\nu)\mathrm{e}^{\mathrm{i}\phi_{\mathrm{i}}(\nu)}}\right]\cancel{\mathrm{e}^{\mathrm{i}\phi_{\mathrm{s}}(\nu)}}}\left[S_{\mathrm{h}}(\nu) - S_{\mathrm{c}}(\nu)\right] + S_{\mathrm{c}}(\nu)\\
&= \frac{\mathcal{F}(I_{\mathrm{s}} - I_{\mathrm{c}})}{\mathcal{F}(I_{\mathrm{h}} - I_{\mathrm{c}})}\left[S_{\mathrm{h}}(\nu) - S_{\mathrm{c}}(\nu)\right] + S_{\mathrm{c}}(\nu)
\end{aligned}
\tag{14}
$$

Since both phase contributions cancel in this expression, the real part should contain the measured signal with noise and the imaginary part only noise. This usually serves as a quality check for the spectra and as a way to estimate the noise level of the data (Knuteson et al., 2004a).

This is the same expression as Eq. 4. Although Revercomb et al. did not consider absorption lines in the black body spectra, everything in Sect. 3, specifically Eq. 11 still holds, so

$$
\begin{aligned}
S_{\mathrm{a,calib}}(\nu) &= \frac{S_{\mathrm{m,s}}(\nu) - S_{\mathrm{m,c}}(\nu)}{S_{\mathrm{m,h}}(\nu) - S_{\mathrm{m,c}}(\nu)}\left[L_{\mathrm{h}}(\nu) - L_{\mathrm{c}}(\nu)\right] + L_{\mathrm{c}}(\nu)\\
&= \frac{\mathcal{F}(I_{\mathrm{s}} - I_{\mathrm{c}})}{\mathcal{F}(I_{\mathrm{h}} - I_{\mathrm{c}})}\left[L_{\mathrm{h}}(\nu) - L_{\mathrm{c}}(\nu)\right] + L_{\mathrm{c}}(\nu)
\end{aligned}
\tag{15}
$$

with $S$ being the complex spectra here.

Unfortunately, it is only true for infinite interferograms that the transmittance cancels in the quotient as we will show in the next section. Next we consider real world, finite interferograms that are truncated and especially unequal-sided interferograms and how absorption lines present in the black body spectra introduce errors.

## 5 Truncated interferograms

So far we have assumed infinite intrinsically asymmetric interferograms from which we get the complex spectrum by the Fourier transform. In the real world all interferograms are finite which can be described by a multiplication of the infinite (potentially asymmetric) interferogram $I(x)$ with a truncation function $A(x)$ (Sakai et al., 1968). Comparing to Eq. 13, the actual measured spectrum is then given by

$$
S_{\mathrm{m}}(\nu) = \int_{-\infty}^{\infty} I(x)A(x)\mathrm{e}^{-\mathrm{i}2\pi\nu x}\,\mathrm{d}\nu = \tilde{I}(\nu) * \tilde{A}(\nu) = \left[r(\nu)\left[S(\nu) + S_{\mathrm{i}}(\nu)\mathrm{e}^{\mathrm{i}\phi_{\mathrm{i}}(\nu)}\right]\mathrm{e}^{\mathrm{i}\phi_{\mathrm{s}}(\nu)}\right] * \tilde{A}(\nu)
\tag{16}
$$

where the tilde over the truncation function and the interferogram denote the Fourier transform. We also call $\tilde{A}$ the scanning function. We can decompose the scanning function into a real and imaginary part, $\tilde{A}_{\mathrm{c}}$ and $\tilde{A}_{\mathrm{s}}$, which are the cosine and sine transforms of the truncation function and arise from the symmetric and antisymmetric parts of the truncation function, respectively:

$$
\tilde{A} = \tilde{A}_{\mathrm{c}} + \mathrm{i}\tilde{A}_{\mathrm{s}}
\tag{17}
$$



We insert this into the quotient of the calibration equation (Eq. 15) to obtain

$$S_{\mathrm{a,calib}} = \frac{\left[ r\left(S_{\mathrm{a}} - L_{\mathrm{c}}\right) T_0 \mathrm{e}^{\mathrm{i}\phi_{\mathrm{s}}} \right] * \tilde{A}}{\left[ r\left(L_{\mathrm{h}} - L_{\mathrm{c}}\right) T_0 \mathrm{e}^{\mathrm{i}\phi_{\mathrm{s}}} \right] * \tilde{A}} \left(L_{\mathrm{h}} - L_{\mathrm{c}}\right) + L_{\mathrm{c}}. \tag{18}$$

If there were no spectral lines from the air between the black bodies and the spectrometer, i. e. $T_0 = 1$, only $S_{\mathrm{a}}$ would be affected by the convolution, since $r$, $L_{\mathrm{bb}}$ and $\phi_{\mathrm{s}}$ are functions that vary slowly with $\nu$ and we can thus pull them out of the convolution with the spectrally narrow function $\tilde{A}$ (Bell, 1972). In the spectral domain this can be understood as the convolution corresponds to a smoothing. With a spectrally narrow smoothing function, high-frequency components will be smoothed, whereas low-frequency components are left unchanged. In the interferogram domain this can be best understood as the interferogram of a spectrum without high-frequency components is concentrated around the ZPD. Truncation (even asymmetric as long as it contains a short section on both sides of the ZPD) does not affect the interferogram. The rest of the terms then cancel and we are left with

$$\begin{aligned} S_{\mathrm{a,calib}} &= \frac{\left[ r\left(S_{\mathrm{a}} - L_{\mathrm{c}}\right) \mathrm{e}^{\mathrm{i}\phi_{\mathrm{s}}} \right] * \tilde{A}}{\left[ r\left(L_{\mathrm{h}} - L_{\mathrm{c}}\right) \mathrm{e}^{\mathrm{i}\phi_{\mathrm{s}}} \right] * \tilde{A}} \left(L_{\mathrm{h}} - L_{\mathrm{c}}\right) + L_{\mathrm{c}} \\ &= \frac{\cancel{r}\left[ \left(S_{\mathrm{a}} * \tilde{A}\right) - \cancel{L_{\mathrm{c}}} \right] \cancel{\mathrm{e}^{\mathrm{i}\phi_{\mathrm{s}}}}}{\cancel{r}\cancel{\left(L_{\mathrm{h}} - L_{\mathrm{c}}\right)}\cancel{\mathrm{e}^{\mathrm{i}\phi_{\mathrm{s}}}}} \cancel{\left(L_{\mathrm{h}} - L_{\mathrm{c}}\right)} + \cancel{L_{\mathrm{c}}} \\ &= S_{\mathrm{a}} * \tilde{A}. \end{aligned} \tag{19}$$

If the truncation is symmetric, this expression is already real, and the imaginary part only contains measurement noise. In the asymmetric case we can take the real part and we are left with

$$S_{\mathrm{a,calib}} = \mathrm{Re}\left[ S_{\mathrm{a}} * \tilde{A} \right] = S_{\mathrm{a}} * \tilde{A}_{\mathrm{c}} \tag{20}$$

which is usually what we expect as the best estimate of the spectrum in FTIR spectroscopy (Sakai et al., 1968). Note that in the asymmetric case, we can now not use the imaginary part anymore for quality control and to estimate the noise level of the data since it does not contain only noise. At the positions of spectral lines, where there are fast intensity variations, the imaginary part shows features similar to those of anomalous dispersion (Sakai et al., 1968).

Now, in case that any spectral lines are present in the spectra used for calibration ($T_0 \neq 1$), we cannot obtain this estimate, since both the numerator and the denominator contain high frequency modes and are thus affected by the convolution. Equation 18 then results in

$$S_{\mathrm{a,calib}} = \frac{\left[\left(S_{\mathrm{a}} - L_{\mathrm{c}}\right) T_0\right] * \left( \tilde{A}_{\mathrm{c}} + \mathrm{i}\tilde{A}_{\mathrm{s}} \right)}{T_0 * \left( \tilde{A}_{\mathrm{c}} + \mathrm{i}\tilde{A}_{\mathrm{s}} \right)} + L_{\mathrm{c}} \tag{21}$$

where for equal-sided truncation $\tilde{A}_{\mathrm{s}} = 0$ and we get

$$S_{\mathrm{a,calib}} = \frac{\left[\left(S_{\mathrm{a}} - L_{\mathrm{c}}\right) T_0\right] * \tilde{A}_{\mathrm{c}}}{T_0 * \tilde{A}_{\mathrm{c}}} + L_{\mathrm{c}}. \tag{22}$$





In order to quantify the effect of the convolutions, we create artificial atmospheric and black body spectra with a line-by-line radiative transfer code, multiply by a smooth artificial phase function, transform them into interferograms, truncate the interferograms and insert them into the Revercomb formula (Eq. 15). We use Arctic atmospheric composition and thermodynamic profiles, since they provide a lower limit to the observed effects because of the low water vapor content in the atmosphere. Due to the low water vapor content, also less lines at the sides of the strong water vapor absorption band are saturated such that these spectral regions might be of use for information retrieval, and thus estimating the errors is important. Furthermore, we operate the NYAEM-FTS in Ny-Ålesund, Svalbard, and a quantitative analysis of errors is needed for the data recorded by this instrument. The deviation from the best estimate (given by Eq. 20) is shown in Fig. 2 for a single line where there is a strong water vapor line such that $T_0$ is much different from 1. The best estimate is shown in blue, the result of applying Revercomb's total power calibration to symmetrically truncated interferograms is shown in orange, and the result of applying the calibration to asymmetrically truncated interferograms is shown in dark gray.

The much larger deviation for asymmetric interferograms can be understood by the additional term $\tilde{A}_s$ in Eq. 21. We cannot simply mitigate this and obtain the estimate of Eq. 22 from unequal-sided interferograms by taking the real part of the numerator and denominator in the Revercomb formula separately, hoping that this makes the contributions of the sine transforms vanish. The phase exponents $\exp(\mathrm{i}\phi_s)$ would not cancel as they do in Eq. 18. The result of this wrong approach is shown in light gray in Fig. 2. Instead, we can determine the phase $\phi_s$ from a small symmetric part around the ZPD of the difference of the interferograms from the hot and cold black body $I_h - I_c$ as is commonly done in phase correction schemes like Mertz' phase correction (Mertz, 1965). A low-resolution phase spectrum is sufficient, since the black body spectra have strong signal to noise everywhere, unlike gas emission spectra, which contain only separate narrow lines. We multiply the inverse of this phase to the numerator and denominator to make them real except for the imaginary part of $\tilde{A}$. Now we can indeed take the real parts of the numerator and denominator separately:

$$
\begin{aligned}
S_{\mathrm{a,calib}} &= \frac{\mathrm{Re}\left[\left[\left[r\left(S_{\mathrm{a}}-L_{\mathrm{c}}\right)T_0\mathrm{e}^{\mathrm{i}\phi_s}\right]*\left(\tilde{A}_{\mathrm{c}}+\mathrm{i}\tilde{A}_{\mathrm{s}}\right)\right]\mathrm{e}^{-\mathrm{i}\phi_s}\right]}{\mathrm{Re}\left[\left[\left[r\left(L_{\mathrm{h}}-L_{\mathrm{c}}\right)T_0\mathrm{e}^{\mathrm{i}\phi_s}\right]*\left(\tilde{A}_{\mathrm{c}}+\mathrm{i}\tilde{A}_{\mathrm{s}}\right)\right]\mathrm{e}^{-\mathrm{i}\phi_s}\right]}\left(L_{\mathrm{h}}-L_{\mathrm{c}}\right)+L_{\mathrm{c}} \\
&= \frac{\mathrm{Re}\left[\left[\left(S_{\mathrm{a}}-L_{\mathrm{c}}\right)T_0\right]*\left(\tilde{A}_{\mathrm{c}}+\mathrm{i}\tilde{A}_{\mathrm{s}}\right)\right]}{\mathrm{Re}\left[\left[\left(L_{\mathrm{h}}-L_{\mathrm{c}}\right)T_0\right]*\left(\tilde{A}_{\mathrm{c}}+\mathrm{i}\tilde{A}_{\mathrm{s}}\right)\right]}\left(L_{\mathrm{h}}-L_{\mathrm{c}}\right)+L_{\mathrm{c}}
\end{aligned}
\tag{23}
$$

which is exactly Eq. 22. Expressed in terms of the measurements we have

$$
\begin{aligned}
S_{\mathrm{a,calib}}(\nu) &= \frac{\mathrm{Re}\left[\left(S_{\mathrm{m,s}}(\nu)-S_{\mathrm{m,c}}(\nu)\right)\mathrm{e}^{-\mathrm{i}\phi_{\mathrm{hc}}}\right]}{\mathrm{Re}\left[\left(S_{\mathrm{m,h}}(\nu)-S_{\mathrm{m,c}}(\nu)\right)\mathrm{e}^{-\mathrm{i}\phi_{\mathrm{hc}}}\right]}\left[L_{\mathrm{h}}(\nu)-L_{\mathrm{c}}(\nu)\right]+L_{\mathrm{c}}(\nu) \\
&= \frac{\mathrm{Re}\left[\mathcal{F}\left(I_{\mathrm{s}}-I_{\mathrm{c}}\right)\mathrm{e}^{-\mathrm{i}\phi_{\mathrm{hc}}}\right]}{\mathrm{Re}\left[\mathcal{F}\left(I_{\mathrm{h}}-I_{\mathrm{c}}\right)\mathrm{e}^{-\mathrm{i}\phi_{\mathrm{hc}}}\right]}\left[L_{\mathrm{h}}(\nu)-L_{\mathrm{c}}(\nu)\right]+L_{\mathrm{c}}(\nu).
\end{aligned}
\tag{24}
$$

with $\phi_{\mathrm{hc}}$ being the phase of the difference interferogram $I_h - I_c$. With this modified version of Revercomb's calibration method, one can now obtain the same spectral estimate from unequal-sided interferograms as from equal-sided interferograms.





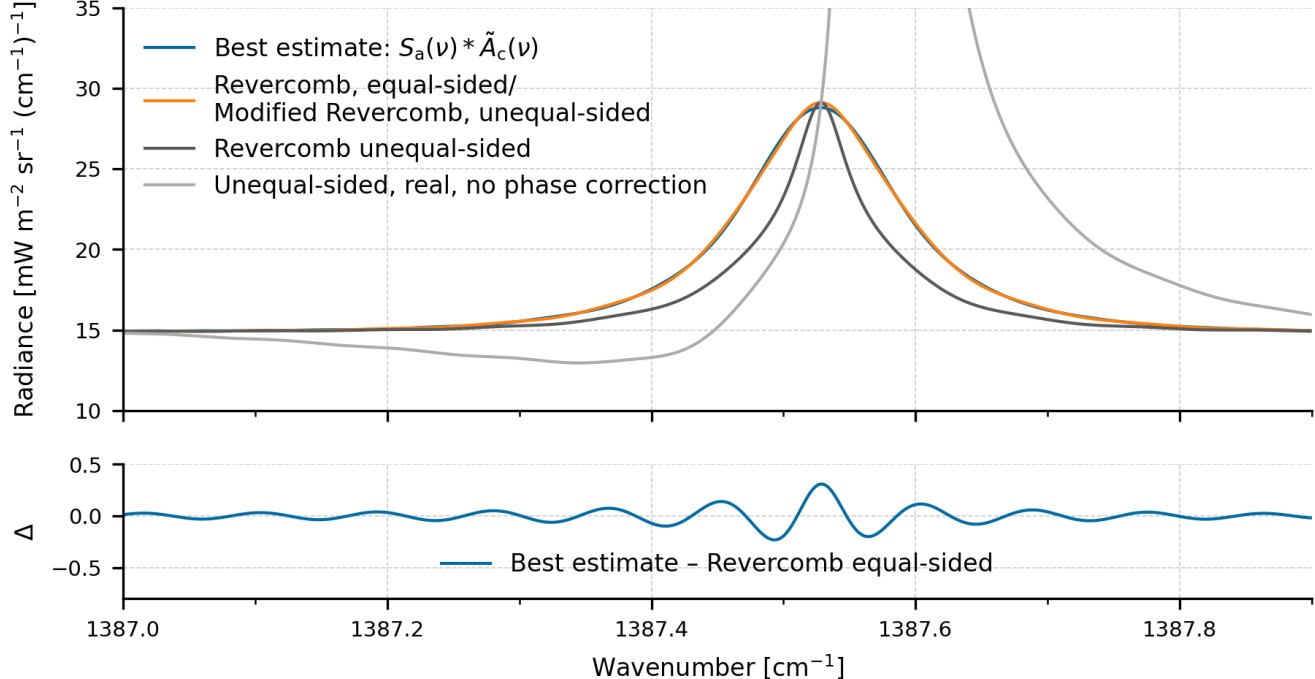

**Figure 2.** Upper panel: The best spectral estimate in FTIR spectroscopy, given by the atmospheric spectrum convolved with the real scanning function from symmetric truncation (Eq. 20), is shown in blue. The spectral estimate from total power calibration, given by either symmetric truncation in the Revercomb formula or equivalently by applying the modified Revercomb calibration to unequal-sided interferograms, is shown in orange. The spectral estimate resulting from applying the complex Revercomb formula to unequal-sided interferograms is shown in dark gray. The estimate resulting from applying the incompletely modified Revercomb formula (real parts only but without phase correction) to unequal-sided interferograms is shown in light gray. Lower panel: Difference between the best estimate without calibration and the estimate with calibration.

Note that these effects of interferogram truncation on the calibrated spectrum depend on the strength of the absorption lines in the black body spectra. Figure 3 shows the differences of the estimates for a larger spectral range than the single line in Fig. 2. In spectral regions where water vapor absorption lines in the black body spectrum are weak, we see that the relative differences generally lie well below 0.05% for equal-sided interferogram truncation or, equivalently, the modified calibration
method with unequal-sided truncation. Even direct application of the Revercomb formula leads to deviations of less than 0.2%. Outside of this region where the effect is much larger, the absorption is so strong however, that in the measured spectrum those lines are saturated in the lowest layer of the atmosphere or, as in the NYAEM-FTS setup, dominated by emission lines from the warm air in the lab (room temperature but typically warmer than outside) between the black bodies and the hatch in the roof. In which case they are of limited use and generally not used for atmospheric retrievals, anyway.





**Figure 3.** As in figure 2, the first panel shows the best estimate in blue, the Revercomb calibration for equal-sided interferograms (modified method for unequal-sided interferograms) in orange and the application of the complex Revercomb method to unequal-sided interferograms in gray, but for a larger part of the spectrum. The second panel shows the relative differences. In the left part, where absorption lines are small in the black body spectrum, the differences are minor and become larger the stronger the absorption lines in the black body spectrum as can be seen on the right hand side (note the different axis scalings). The black body spectra are shown in the bottom panel. All are synthetic spectra from a realistic Arctic atmospheric profile.





## 6    Conclusions

We have investigated total power calibration taking interferogram truncation and absorption lines in the black body spectra into account. We have found that because of the convolution with the scanning function, we cannot obtain the same spectral estimate of the atmospheric spectrum after calibration, which is commonly referred to as the best estimate in FTIR spectroscopy. We have described the underlying theory and have shown that while the application of the Revercomb formula to unequal-sided interferograms might lead to even larger deviations from the best estimate than the application to equal-sided interferograms, one can always obtain the same estimate as for equal-sided interferograms by modifying the calibration method incorporating a phase correction step. We have quantified the deviations from the best estimate in a case study with simulations of realistic spectra from a line-by-line radiative transfer model. We have used a representative state of the Arctic atmosphere in Ny-Ålesund since the Arctic atmosphere with its low water vapor content provides a lower limit of the error introduced by water vapor lines in the black body spectra, and quantitative error analysis is needed for the NYAEM-FTS. While the errors might be large in regions with strong absorption lines, the atmospheric spectra are also saturated in those regions from just the lowermost layer. The useful information in those spectral regions is thus also limited, and most of the time we can and usually should exclude those regions from retrieval procedures. In the other spectral regions the errors lie well below 0.05 % in our exemplary application.

*Code and data availability.*   The data used in the plots of this paper are available in the supplement. The code is available on request.

*Author contributions.*   L.H. conceived of the presented idea, developed the theory, carried out the analysis and prepared the manuscript. M.B. supervised the project, discussed the analysis and contributed to the manuscript. M.P. and J.N. commented on the manuscript.

*Competing interests.*   The authors declare that they have no conflict of interest. At least one of the coauthors is a member of the editorial board of AMT.

*Acknowledgements.*   L.H. thanks Anne Kleinert for insightful discussions on calibration within FTIR emission spectroscopy and the hint that Revercomb's total power calibration is only applicable to equal-sided interferograms. The radiative transfer equations as well as the Python implementation of the line-by-line code come mainly from ChatGPT Model 4o. This includes the equations for a layered black body scene in Sect 3. The radiative transfer code was validated against SFIT4, a standard radiative transfer and trace gas retrieval code used by the NDACC Infrared Working Group. ChatGPT Model 4o also contributed the result for the convolution of constant and slowly varying functions with spectrally narrow functions such as the Fourier transform of a (shifted) boxcar function. We thankfully acknowledge the AWIPEV station teams, who continuously provide on-site support for our project with grant number AWIPEV_0004. We thankfully acknowledge funding



from the University of Bremen through the junior research group "Greenhouse gases in the Arctic" and the German Research Foundation (DFG) – Projektnummer 268020496 – TRR 172 – (AC)3.



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
