# Peer review of "Total power calibration in FTIR emission spectroscopy for finite interferograms"

_EGUsphere, 2025_

## Referee Comment (RC1)

**Referee comments for "Total power calibration in FTIR emission spectroscopy for finite interferograms" by Heizmann et al., (2025)**

**1   General Comments**

This paper presents improvements to FTIR emission spectroradiometery by modifying the standard total power calibration procedure by Revercomb et al., (1988), to improve the robustness and capability of such instruments when measuring unequal-sided interferograms by correcting the phase error through deriving a correction factor from the reference blackbodies. This work is within scope for AMT, has sound methodology, and I believe it may be published after the following revisions have been addressed.

In my opinion, the most significant improvement which may be made is that this work does not adequately present the scope of this work, the wider implications of this calibration technique, and how this new technique compares to the usual method. The introduction, in its current state, is brief and does not adequately introduce the utility of the work presented. The number of references to primary sources is also lacking, and some suggestions are provided in the following comments. Two instruments are introduced, the AERI and NYAEM-FTS, but measurements from either of these instruments are not strongly featured in the discussion. Given that the authors have specified improvements to trace gas retrievals as one of the motivating factors for higher resolution emission FTIR spectra, there should be some introduction on exactly why current instrument resolution (e.g., AERI) is insufficient, and why higher resolution spectra would improve retrievals.

It would also make sense for Section 2 to be moved to the introduction, as that is background material, and for the main results (that is, Figures 2 and 3 and associated text) should be expanded, possibly showcasing wider applicably of this calibration method. Specific comments and technical suggestions are given below.

**2   Specific Comments**

Lines 17-18: Emission FTIR instruments have been used to retrieve cloud microphysical properties (e.g., Turner et al., 2005; Rowe et al., 2019; Richter et al., 2022), any of which would be a better reference for cloud radiative properties.

Turner and Blumberg (2019) also use AERI spectra for thermodynamic profiling of the atmosphere. Along that line, an updated (and much longer time series of) AERI trace gas retrievals is available at Hung et al., (2025). You could point to the limited AERI spectral resolution being one of the limiting factors for vertical sensitivity to trace gases in that work to motivate the need for improvements to emission FTIR spectral resolution for trace gas measurements. The Mariani et al., (2013) reference retrieves column amounts of gases, and so the retrieval may be (presumably) less sensitive to spectral resolution.

Line 21: "In order to achieve higher resolution, which is especially important for trace gas retrievals" should include a reference. There are many suitable references among the solar FTIR community, such as through NDACC-IRWG or TCCON, which discuss fitting various line broadening effects. I would also suggest adding some additional text to motivate why one would want to use (specifically) FTIR emission spectroscopy to measure atmosphere composition (e.g., do not need sunlight, better equipped to monitor diurnal cycles, workable during polar night, etc).

Line 25: "trace gas retrievals are currently being developed". Is it possible to include an example of this in your results? I appreciate that this may be a significant amount of effort, so I do not believe that this is mandatory for this work, but I believe it would strengthen this paper to include an example/case study of such a retrieval (possibly on simulated spectra) when this new calibration technique is applied, especially for gases which have retrieval windows coinciding with some water lines.

Line 27: "most information" is ambiguous. Does "information" here refer to the spectra or retrievals of interesting quantities from the spectra such as the previously mentioned trace gases or aerosols.

Figure 1: Please define the y-axis unit "a. u.".

Lines 105-106: This sentence is confusing. If I am reading it correctly, it may be rephrased as "...i.e., there is a phase shift $\phi_i(\nu)$ relative to the source signal (external) $\phi_s(\nu)$ and therefore the amplitude of the spectra. This means the spectra cannot be used because of artifacts due to the complex contribution". Please clarify accordingly.

Lines 162-163: Provide some more detail on "We use Arctic atmospheric composition and thermodynamic profiles". For example, specifying the precipitable water vapour is important for putting your simulated spectra in context. Depending on your location, the Arctic humidity can vary one or two orders of magnitude between summer and winter, so one spectra would likely not be representative of the range of conditions found in the region. It may also be useful to look at two spectra to capture this range, one for low humidity (e.g., 0.1 cm precipitable water) and one for high humidity (e.g., 2 cm).

Figures 2 and 3: Given that one of your main motivations is to improve spectral line fitting for trace gas retrievals, including similar plots for the windows used to perform those retrievals is important (e.g., the windows used for AERI retrievals in the Mariani et al. (2013) and Hung et al. (2025) references), especially when those windows contain some water lines.

Lines 186-194: My understanding is that the particular window in Fig. 2 was chosen due to the strong water line, where the modified calibration should provide the best performance gain. The authors say the relative differences are much less when the water vapor absorption is weak - it would be good to quantify this by including the exact numbers for spectral regions of interest. Again, windows used for trace gas retrievals are suggested, especially for evaluating suitability of higher resolution spectra for such retrievals.

Lines 202-205: Add the exact quantification for specific claims in the conclusion. E.g., for the statements "We have quantified the deviations from the best estimate..." and "...provides a lower limit of the error introduced by water vapor lines...".

**3   Technical Corrections**

Instances of "i. e." and "e. g." in the text should be changed to "i.e., and e.g.," throughout the text.

Line 55 and Equation 1: Define $\nu$, or specify that this (and following) equations are for monochromatic light.

Line 47: Change to "Most applications, e.g., trace gas retrievals, ...".

Line 92: Remove "basically".

Line 97: Insert a comma after "In Fourier transform spectroscopy".

Line 103: Change to "This phenomenon is referred to as a phase error..".

Line 110: The reference "Revercomb et al." is missing the year.

Line 118: Same as above.

Line 121: Remove "here".

Line 127: Insert comma after "In the real world all interferograms are finite"

Line 143: Change to "spectrum without high-frequency components being concentrated around the ZPD".

Line 172: replace "mitigate" with "ignore".

Line 176: Do you mean "Mertz's".

Line 194: remove ", anyway".

Figure 3 caption: capitalize "Figure 2".

Line 207: Add commas to the following text "most of the time we can, and usually should,".

**4   References**

Hung, J., Liu, L., Palm, M., Mariani, Z., Manney, G. L., Millán, L. F., Strong, K. (2025). Autonomous Year-Round Measurements of $O_3$, CO, $CH_4$, and $N_2O$ in the High Arctic With the Atmospheric Emitted Radiance Interferometer. Journal of Geophysical Research: Atmospheres, 130(11). https://doi.org/10.1029/2024JD042847

Richter, P., Palm, M., Weinzierl, C., Griesche, H., Rowe, P. M., Notholt, J. (2022). A dataset of microphysical cloud parameters, retrieved from Fourier-transform infrared (FTIR) emission spectra measured in Arctic summer 2017. Earth System Science Data, 14(6), 2767–2784. https://doi.org/10.5194/essd-14-2767-2022

Rowe, P. M., Cox, C. J., Neshyba, S., Walden, V. P. (2019). Toward autonomous surface-based infrared remote sensing of polar clouds: Retrievals of cloud optical and microphysical properties. Atmospheric Measurement Techniques, 12(9), 5071–5086. https://doi.org/10.5194/amt-12-5071-2019

Turner, D. D. (2005). Arctic Mixed-Phase Cloud Properties from AERI Lidar Observations: Algorithm and Results from SHEBA. Journal of Applied Meteorology and Climatology, 44(4), 427–444. https://doi.org/10.1175/JAM2208.1

Turner, D. D., Blumberg, W. G. (2019). Improvements to the AERIoe Thermodynamic Profile Retrieval Algorithm. IEEE Journal of Selected Topics in Applied Earth Observations and Remote Sensing, 12(5), 1339–1354. https://doi.org/10.1109/JSTAR

---

## Author Comment (AC1)

**Response to Referee Comments**

Manuscript title: *Total power calibration in FTIR emission spectroscopy for finite interferograms*

Lukas Heizmann, Mathias Palm, Justus Notholt, and Matthias Buschmann

We thank the reviewer for their careful reading of our manuscript and for the valuable comments and suggestions. In the following, we address each point raised.

**Reviewer 1 Comments and Responses**

**Comment:** In my opinion, the most significant improvement which may be made is that this work does not adequately present the scope of this work, the wider implications of this calibration technique, and how this new technique compares to the usual method. The introduction, in its current state, is brief and does not adequately introduce the utility of the work presented.

**Response:** We acknowledge that the manuscript in its original state is too brief and we extend the introduction and the conclusion in the revised version in order to more clearly present the scope of this work and its implications on the application of total power calibration in FTIR emission spectroscopy.

We now emphasize more explicitly that there is an incompleteness in the paper by Revercomb et al. (1988) as the finiteness of real-world interferograms and its implications on the total power calibration are not considered. As such, the main motivation for this work is theoretical in nature at first but in principle relevant for all real-world measurements where Revercomb's total power calibration method is applied.

We explain more clearly the line of thought of our work: We present the mathematical description of this issue. We show that the problem of unequal-sided interferogram truncation can always be reduced to the problem of equal-sided interferogram truncation by means of a phase correction step within the calibration routine. This is important because the error introduced by the application of total power calibration with unequal-sided finite interferograms is generally larger than the error introduced by equal-sided finite interferograms. This means that the usage of unequal-sided interferograms to obtain higher spectral resolution poses no greater fundamental limitation than that of equal-sided interferograms of the same resolution. As an example, we perform a case study for the NYAEM-FTS which is operated by our group in Ny-Ålesund. We present quantitative differences between the optimal estimate on the spectrum from truncated interferograms as described by Sakai et al. (1968), the original Revercomb method applied to unequal-sided interferograms and the new method including a phase correction step, which is equivalent to applying the standard method to equal-sided interferograms. In our case, we find that errors are small in spectral regions of interest for trace gas retrievals. This result is reassuring for our application and might be generalizable to others, as major effects occur where water vapor or carbon dioxide absorption lines are very strong. In those spectral regions, however, the information content on the atmospheric state above the lowest layer is strongly limited because of saturation effects and is therefore of limited practical use. However, the quantitative relevance of the limitations of total power calibration with finite interferograms needs to be investigated for each application individually, since it depends on the instrument, the measurement site, and the intended use of the spectra. We therefore recommend in the revised conclusion that such

an analysis be performed for each individual use case in order to estimate the radiance error arising from finite interferograms, to decide on the necessity of applying the additional phase correction step for unequal-sided interferograms, and, for example, to inform retrieval window selection.

**Comment:** The number of references to primary sources is also lacking, and some suggestions are provided in the following comments.

**Response:** We thank the reviewer for these suggestions and provide the references in the revised manuscript.

**Comment:** Two instruments are introduced, the AERI and NYAEM-FTS, but measurements from either of these instruments are not strongly featured in the discussion.

**Response:** We agree and would like to emphasize that this paper is more conceptual in nature. The analysis of the NYAEM-FTS serves as an illustrative example and to assess the impact for our instrument deployed in Ny-Ålesund. But in general, we recommend that such an analysis be performed for each individual use case.

**Comment:** Given that the authors have specified improvements to trace gas retrievals as one of the motivating factors for higher resolution emission FTIR spectra, there should be some introduction on exactly why current instrument resolution (e.g., AERI) is insufficient, and why higher resolution spectra would improve retrievals.

**Response:** We agree and now cite Hung et al. (2025), which demonstrates limited vertical resolution in profile trace gas retrievals from current resolution FTIR emission spectra. We explain that the vertical profile resolution can be expected to increase with increasing spectral resolution as more information on the line shape increases the sensitivity of the retrieval to different altitudes along the observed light path.

**Comment:** It would also make sense for Section 2 to be moved to the introduction, as that is background material, and for the main results (that is, Figures 2 and 3 and associated text) should be expanded, possibly showcasing wider applicably of this calibration method. Specific comments and technical suggestions are given below.

**Response:** We move the content of Section 2 into the introduction to clearly separate the background material from the new developments in this work.

**Comment:** Lines 17-18: Emission FTIR instruments have been used to retrieve cloud microphysical properties (e.g., Turner et al., 2005; Rowe et al., 2019; Richter et al., 2022), any of which would be a better reference for cloud radiative properties. Turner and Blumberg (2019) also use AERI spectra for thermodynamic profiling of the atmosphere. Along that line, an updated (and much longer time series of) AERI trace gas retrievals is available at Hung et al., (2025).

**Response:** We thank the reviewer for these helpful suggestions and will include the references in the revised manuscript.

**Comment:** You could point to the limited AERI spectral resolution being one of the limiting factors for vertical sensitivity to trace gases in that work to motivate the need for improvements to emission FTIR spectral resolution for trace gas measurements. The Mariani et al., (2013) reference retrieves column amounts of gases, and so the retrieval may be (presumably) less sensitive to spectral resolution.

**Response:** We strongly agree and make this point in the revised manuscript.

**Comment:** Line 21: "In order to achieve higher resolution, which is especially important for trace gas retrievals" should include a reference. There are many suitable references among the solar FTIR community, such as through NDACC-IRWG or TCCON, which discuss fitting various line broadening effects.

**Response:** We agree and provide such a reference in the revised manuscript.

**Comment:** I would also suggest adding some additional text to motivate why one would want to use (specifically) FTIR emission spectroscopy to measure atmosphere composition (e.g., do not need sunlight, better equipped to monitor diurnal cycles, workable during polar night, etc).

**Response:** We agree and add such a comment in the manuscript.

**Comment:** Line 25: "trace gas retrievals are currently being developed". Is it possible to include an example of this in your results? I appreciate that this may be a significant amount of effort, so I do not believe that this is mandatory for this work, but I believe it would strengthen this paper to include an example/case study of such a retrieval (possibly on simulated spectra) when this new calibration technique is applied, especially for gases which have retrieval windows coinciding with some water lines.

**Response:** We agree that this would strengthen the paper but consider this beyond the scope of this work and more appropriate for a future study on trace gas retrievals with the NYAEM-FTS.

**Comment:** Line 27: "most information" is ambiguous. Does "information" here refer to the spectra or retrievals of interesting quantities from the spectra such as the previously mentioned trace gases or aerosols.

**Response:** We were referring to most of the interesting quantities that can be retrieved from the spectra and clarify this in the revised manuscript.

**Comment:** Figure 1: Please define the y-axis unit "a. u.".

**Response:** We chose "radiance in arbitrary units" since the quantity shown is technically the Fourier transform of the electric signal detected after the interferometer. This corresponds to radiance but is not given in the usual units of radiance $(\mathrm{W}\,\mathrm{m}^{-2}\,\mathrm{sr}^{-1}\,(\mathrm{cm}^{-1})^{-1})$ because of the (wavenumber dependent) responsivity of the instrument and the possible offset from internal emission. The spectrum is given in the usual radiance units only after performing the total power calibration.

**Comment:** Lines 105-106: This sentence is confusing. If I am reading it correctly, it may be rephrased as "...i.e., there is a phase shift $\phi_i(\nu)$ relative to the source signal (external) $\phi_s(\nu)$ and therefore the amplitude of the spectra. This means the spectra cannot be used because of artifacts due to the complex contribution". Please clarify accordingly.

**Response:** This is not what we meant to say, but rather: "...i.e., there is a phase shift $\phi_i(\nu)$ relative to the source signal (external) $\phi_s(\nu)$. In the absence of such a phase shift, we can obtain the measured real spectrum (responsivity times the sum of the source spectrum and the internal emission) from the magnitude of the complex signal. However, this is not possible if a phase shift is present because cross terms arise."

**Comment:** Lines 162-163: Provide some more detail on "We use Arctic atmospheric composition and thermodynamic profiles". For example, specifying the precipitable water vapour

is important for putting your simulated spectra in context. Depending on your location, the Arctic humidity can vary one or two orders of magnitude between summer and winter, so one spectra would likely not be representative of the range of conditions found in the region. It may also be useful to look at two spectra to capture this range, one for low humidity (e.g., 0.1 cm precipitable water) and one for high humidity (e.g., 2 cm).

**Response:** We agree and provide two spectra, representative of summer and winter conditions in Ny-Ålesund with substantial differences in precipitable water vapor, and provide the corresponding atmospheric profiles.

**Comment:** Figures 2 and 3: Given that one of your main motivations is to improve spectral line fitting for trace gas retrievals, including similar plots for the windows used to perform those retrievals is important (e.g., the windows used for AERI retrievals in the Mariani et al. (2013) and Hung et al. (2025) references), especially when those windows contain some water lines.

**Response:** We provide the entire mid-infrared spectral range and evaluate particularly the windows used for trace gas retrievals in the mentioned publications.

**Comment:** Lines 186-194: My understanding is that the particular window in Fig. 2 was chosen due to the strong water line, where the modified calibration should provide the best performance gain. The authors say the relative differences are much less when the water vapor absorption is weak - it would be good to quantify this by including the exact numbers for spectral regions of interest. Again, windows used for trace gas retrievals are suggested, especially for evaluating suitability of higher resolution spectra for such retrievals.

**Response:** We add quantitative values for the spectral regions of interest in the revised version of the manuscript.

**Comment:** Lines 202-205: Add the exact quantification for specific claims in the conclusion. E.g., for the statements "We have quantified the deviations from the best estimate..." and "...provides a lower limit of the error introduced by water vapor lines...".

**Response:** We add quantitative values to the revised conclusions.

The technical corrections are applied in the revised manuscript.

**Acknowledgments**

ChatGPT 5.2 was used to assist with the author responses to improve use of language, clarity, and structure of the argumentation. The original text before AI assistance and the AI responses are available upon request.